

# Association of toll-like receptors single nucleotide polymorphisms with HBV and HCV infection: research status

Yaxin Xu[*], Wentao Xue[*], Hongwei Gao, Jiabo Cui, Lingzhi Zhao and Chongge You

Laboratory Medicine Center, Lanzhou University Second Hospital, Lanzhou City, Gansu Province, China
[*] These authors contributed equally to this work.

## ABSTRACT

**Background**. Hepatitis B virus (HBV) and hepatitis C virus (HCV) infections have become increasingly severe worldwide and are a threat to public health. There have been a number of studies conducted recently on the relationship of single nucleotide polymorphisms (SNPs) to innate immune receptor genes such as toll-like receptors (TLRs). Some literature suggests that SNPs of TLRs are associated with HBV and HCV infection. We summarized the role of *TLRs* gene polymorphisms associated with HBV and HCV infections and explored their possible mechanisms of action.

**Methodology**. PubMed and Web of Science were used to perform the literature review. Related articles and references were identified and used to analyze the role of *TLRs* gene polymorphism in HBV and HCV infection.

**Results**. *TLRs* gene polymorphisms may have beneficial or detrimental effects in HBV and HCV infection, and some SNPs can affect disease progression or prognosis. They affect the disease state by altering gene expression or protein synthesis; however, the mechanism of action is not clearly understood.

**Conclusions**. Single nucleotide polymorphisms of TLRs play a role in HBV and HCV infection, but the mechanism of action still needs to be explored in future studies.

## INTRODUCTION

Infections by the hepatitis B virus (HBV) and hepatitis C virus (HCV) are two major growing health problems. Over 2 billion people are infected with HBV and 210 million people in the world are infected with HCV. Among them, more than 290 million people develop chronic HBV infections (*Helbig & Beard, 2012*; *Shepard, Finelli & Alter, 2005*; *Collaborators, 2018*). These two viral infections are the leading causes of chronic hepatitis (CH), liver cirrhosis (LC), and hepatocellular carcinoma (HCC).

The progression of chronic hepatitis B and C is strongly influenced by host, virus and environmental factors (*Lingala & Ghany, 2015*; *An, 2018*; *Zeng, 2014*). Susceptibility to viral infection and the progression of related diseases result from the interaction of host and viral genetic characteristics mediated by environmental, physiological and metabolic factors (*Ellwanger et al. , 2018*). *Han et al. (2013)* found that the effect of miRNA SNPs

Corresponding author
Chongge You, youchg@lzu.edu.cn

on the susceptibility to HBV-related HCC may be greatly affected by HBV mutations. Therefore, host genetic polymorphisms may be associated with infections by a specific HBV genotype. In this respect, conflicting results in studies evaluating the same specific host polymorphism in the context of HBV-related diseases may be due to differences in the ethnic background of the studied population or by circulating HBV genetic variations (*Ellwanger et al., 2018*). Additionally, environmental factors, stress, nutrition, and exposure to different pathogens may affect the immune response and gene expression, which cannot be ignored in genetic research (*Ellwanger et al. , 2018*). Taking together, this can explain why the same SNP has different effects in different populations. For example, in the TLR3 rs3775290 C/T polymorphism the T allele was slightly associated with chronic HBV infection in the Tunisian population (*Sghaier et al., 2019*), while the TT genotype was a protective factor in the Chinese population (*Huang et al., 2015*). In addition, the T allele was reported to be a risk factor for chronic HCV infection in the Tunisian population and the C allele was a protective factor (*Sghaier et al., 2019*; *Mosaad et al., 2019*), while the CC genotype was found to be a risk factor for chronic HCV infection in an Egyptian cohort (*Hamdy et al., 2018*). This contradictory result reinforces the fact that, in addition to host genetic characteristics, viral genetic characteristics and environmental factors also play a very important role in viral susceptibility.

Toll-like receptors are a large class of proteins and are expressed in all innate immune cells, including macrophages, dendritic cells (DCs), neutrophils, natural killer (NK) cells, etc. They are the basis of the innate immune response and are necessary for an effective defense against host-virus infection (*El-Zayat, Sibaii & Mannaa, 2019*; *Delneste, Beauvillain & Jeannin, 2007*). They recognize different molecular motifs, called pathogen-associated molecular patterns (PAMPs) and damage-associated molecular patterns (DAMPs) (*Mann, 2011*; *Woo, Corrales & Gajewski, 2015*). To date, 10 functional TLRs have been identified in human beings (*Mann, 2011*). According to their subcellular location, they can be divided into two groups. One group is expressed on the cell surface and includes TLR1, 2, 4-6 and 10. The second group is located on the membranes of intracellular compartments (such as endosomes, lysosomes, endolysosomes and endoplasmic reticulum), including TLR3 and 7-9, which allows them to recognize the nucleic acid of invasive viruses and induce pro-inflammatory responses to resist viral infection (*Kemball, Alirezaei & Whitton, 2010*; *Feng & Chao, 2011*; *Topping & Kelly, 2019*). The activation of TLRs results in the recruitment of myeloid differentiation primary response 88 (MyD88), Toll/IL-1 receptor domain-containing adaptor protein (TIRAP), Toll/IL-1 receptor domain-containing adaptor protein inducing interferon (IFN)-β (TRIF), and TRIF-related adapter molecules (TRAM) in the cytoplasm. Two major adaptor pathways, the MyD88 and TRIF pathways, have been observed in TLRs signal. MyD88 can directly bind to the cytoplasmic domain and induce gene expression through the transcription factors nuclear factor kappa-light-chain-enhancer of activated B cells (NF-κB), activator protein (AP)-1 or interferon response factor (IRF)-1, IRF-5 and IRF-7 (*Chen, Szodoray & Zeher, 2016*). The activation of MyD88-NF-κB leads to the production of various pro-inflammatory cytokines, including TNF-α and IL-6 (*Takeuchi & Akira, 2010*; *Yamamoto & Takeda, 2010*), whereas the TRIF pathway leads to the activation of transcription factors such as IRF-3 and NF-kB, and induces the

production of IFN-β and inflammatory cytokines (*Vercammen, Staal & Beyaert, 2008*). They conduct signals to activate bactericidal and pro-inflammatory responses, thereby clearing infectious agents (*Shigeoka et al., 2007*; *Yamamoto et al., 2003*; *Yamamoto et al., 2002*; *Bortoluci & Medzhitov, 2010*) (Fig. 1).

There is increasing evidence that TLRs play an important role in the pathophysiology of a variety of liver diseases, possibly because TLRs are widely expressed in all types of liver cells, including hepatocytes, Kupffer cells (KCs), hepatic stellate cells (HSC), and immune cells such as hepatic dendritic cells (*Roh & Seki, 2013*). The inflammatory signaling pathways mediated by TLRs are associated with a series of liver diseases, such as hepatitis, liver fibrosis, liver cirrhosis, alcoholic and non-alcoholic liver disease, ischemia/reperfusion injury, liver regeneration and hepatocellular carcinoma (*Roh & Seki, 2013*; *Chen & Sun, 2011*). The activation of TLR2 may lead to liver injury in chronic HCV infection (*Nelson, 2001*). TLR2 associates with TLR1 and TLR6 and plays a key role in the recognition of HCV core and NS3 proteins by innate immune cells and induction of TNF-α, IL-1β, IL-6, IL-8 and IL-10 production (*Szabo & Dolganiuc, 2005*). Furthermore, TLR3 activation induced acute liver injury has been observed in HBV transgenic mice (*Chen et al., 2007a*; *Chen et al., 2007b*). The activation of TLR3 significantly enhanced the IFN-γ receptor expression in HBS-B6 mice hepatocytes, and thus significantly increased the activation of downstream signal phosphorylation transducers and activators of transcription 1-interferon regulatory factor-1 (pSTAT1-IRF-1) (*Chen et al., 2007b*). It has also been found that TLR4 and intestinal microbiota play a role in HCC promotion, mediating increased proliferation, production of pro-inflammatory cytokines (TNF-α and IL-6), expression of the hepatomitogen epiregulin and preventing apoptosis (*Dapito et al., 2012*; *Yu et al., 2010*). Mice deficient in TLR4 and MyD88 showed a decreased in the incidence, size, and number of chemical-induced liver cancers, suggesting that TLRs signaling contribute significantly to hepatocarcinogenesis (*Seki & Brenner, 2008*; *Naugler et al., 2007*). Therefore, it is necessary to study the relationship between genetic polymorphisms in TLRs and HBV and HCV infection progression.

Single nucleotide polymorphisms (SNPs) are the most common form of genetic variation in the human genome and some of them may play a role in the susceptibility to disease. Recently, studies have found that TLRs gene polymorphisms play a significant role in disease susceptibility (*Schröder & Schumann, 2005*; *Misch & Hawn, 2008*; *El-Omar, Ng & Hold, 2008*). SNPs in TLR genes may lead to changes in protein or gene expression, thereby affecting the function and efficacy of signal transduction, altering the immune response, and affecting the susceptibility, progression and treatment of diseases. For example, the TLR2 rs4696480 polymorphism is located in the promoter region, which we believe may partially inhibit pro-inflammatory cytokines by reducing TLR2 transcription, and thus may have the potential to reduce inflammatory damage in chronic HBV infection (*Lin et al., 2018*). The TLR3 rs3775290 polymorphism changes the extracellular domain of TLR3 and damages the function of receptor. This may inactivate TLR3 signal during virus infection (*Sghaier et al., 2019*; *Gao et al., 2015*), resulting in reduced IRF-3 activation and type I IFN production (*Zhang et al., 2007*), thus causing a recognition disorder of pathogenic microorganisms and inadequate immune response (*Geng et al., 2016*).
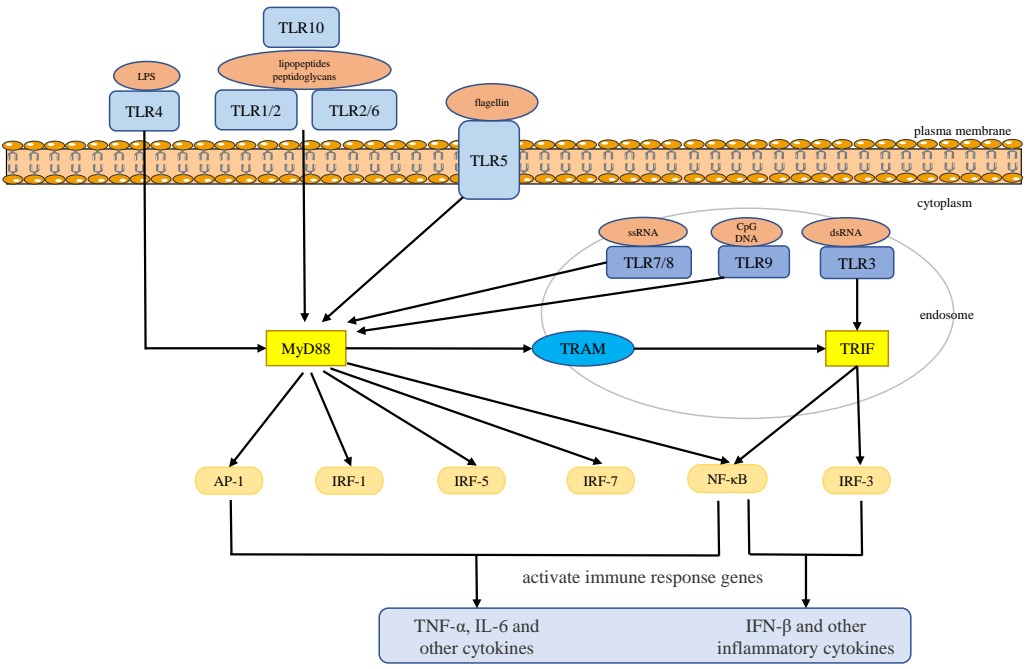

**Figure 1  Toll-like receptor function and signal transduction.** The 10 toll-like receptors (TLRs) are divided into extracellular (TLR1, TLR2, TLR4, TLR5, TLR6 and TLR10) and intracellular (TLR3, TLR7, TLR8 and TLR9) subtypes. Different components of microbial activate the extracellular TLRs and the intracellular TLR3, TLR7/TLR8, and TLR9 recognize viral dsRNA, ssRNA and unmethylated CpG DNA, respectively. MyD88 and TRIF are the two main junction pathways in TLR signal transduction. They transmit the signal downstream and activate immune response genes *via* transcription factors such as NF-κ B, AP-1, or IRF-1, IRF-3, IRF-5, and IRF-7 and induce the production of cytokines such as TNF-α and IFN-β.

The purpose of this review is to summarize the role of TLRs polymorphisms in HBV and HCV infection and their impact on the progression and treatment of HBV and HCV-related liver disease (CH, LC, and HCC). At the same time, we discussed the possible mechanism of action of some SNPs, so as to understand the current research status.

## SURVEY METHODOLOGY

### Search strategy

A complete electronic search was conducted in PubMed and Web of Science databases to collect English-language publications (January 2010 to January 2022). The objects of study were limited to human beings and the search terms included "Toll-like receptor", "TLR", "single nucleotide polymorphism", "SNP", "polymorphism", "genetic variant", "hepatitis B", "HBV", " hepatitis C", "HCV", and related MeSH terms. These search terms are combined and retrieved.

### Inclusion criteria
• Human beings as the object of study.

- Studies on the relationship between TLRs polymorphisms and the risk of HBV and HCV infection.
- Studies on the effect of TLRs polymorphisms on the progression and treatment of HBV and HCV-related liver diseases.
- Only published full-text studies were included.

### Exclusion criteria

- Studies that do not meet the current inclusion criteria.
- Studies that are irrelevant to the topic.
- Studies that contain only abstracts but no full text in the database, and they are not available through other means.
- Reviews, editorials, commentary, abstracts with insufficient data, and conference proceedings (Fig. 2).

## SNPS OF TOLL-LIKE RECEPTORS ASSOCIATED WITH HBV

### TLR2 gene

The included studies found three SNPs associated with HBV in the *TLR2* gene, including rs3804100, rs3804099 and rs4696480 (Table S1). *Chen et al. (2017)* suggested that the rs3804100 polymorphism may be a protective factor for HBV-related HCC in the Chinese male population. Among non-drinkers, the risk of HBV-related HCC decreased in the CT and CC genotype populations. Furthermore, the polymorphism was significantly associated with the status of vaccine-induced serum anti-HBV response, and the TT genotype was beneficial to the production of an anti-HBV response (*Chen et al., 2011*). It has been reported that the rs3804099 CT and TT genotypes had a protective effect on the progression of hepatitis B (*Lin et al., 2018*; *Junjie et al., 2012*). In addition, Xie et al. (*Junjie et al., 2012*) performed haplotype analysis on the two locus rs3804100- rs3804099. They found that the haplotype TT was significantly associated with a reduction in the risk of HCC. In contrast, the haplotype CC significantly increased the risk of HCC in patients. For the rs4696480 polymorphism, it has been shown that the TT genotype significantly reduced the odds of hepatitis B progression (*Lin et al., 2018*).

A study has shown that HBV induced the inflammatory response of monocytes through the TLR2/MyD88/NF-κB signaling pathway. In patients with chronic hepatitis B, the mRNA expression levels of IL-1β, TNF-α and IL-10 were significantly higher than in healthy individuals (*Song et al., 2019*). *Lin et al. (2018)* found that the rs3804099 and rs4696480 polymorphisms were associated with inhibiting IL-6 and TNF-α levels. The rs4696480 is located in the promoter region, which we believe may partially inhibit pro-inflammatory cytokines by reducing the transcription of *TLR2*. Those who carry the two polymorphisms showed an improvement in liver function parameters (ALT, AST, etc.), indicating that they may have the potential to reduce inflammatory damage in chronic HBV infection.

### *TLR3* gene

There are seven SNPs associated with HBV in the *TLR3* gene (Table 1). A study in a Saudi Arabian cohort found that the T allele of the rs1879026 polymorphism was significantly

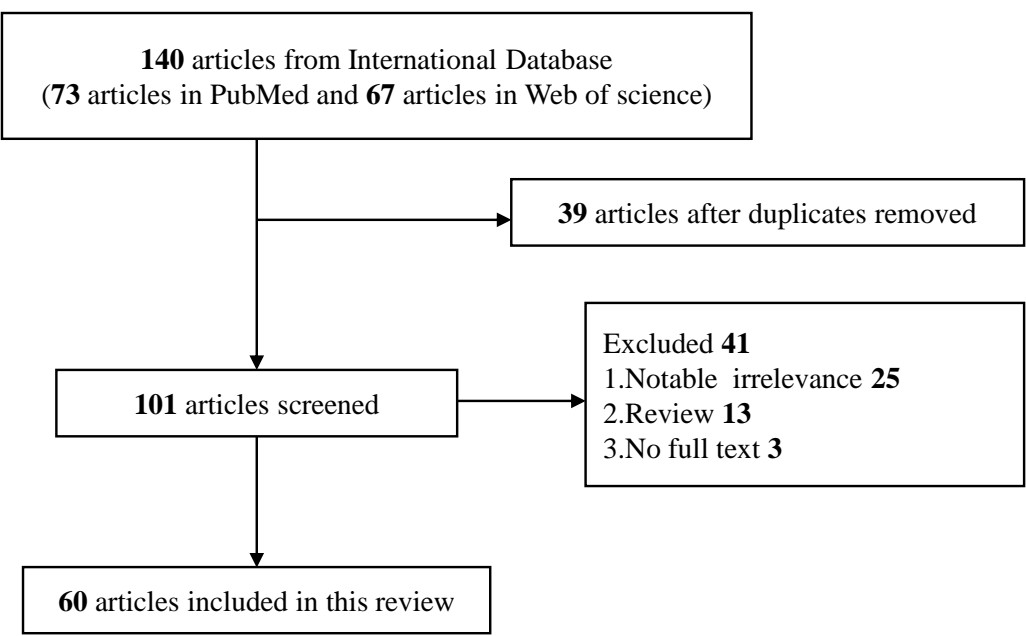

**Figure 2  Flow diagram of selection for eligible studies enrolled in this review.**

associated with the risk of HBV infection (*Al-Qahtani et al., 2012*). However, a lack of association between the polymorphism and HBV-related liver disease was found in a Chinese cohort (*Huang et al., 2015*). This suggests that the rs1879026 polymorphism may have ethnic differences, or the polymorphism may only affect HBV susceptibility, but not HBV-related liver diseases. In haplotype analysis, the haplotype GCGA of rs1879026-rs5743313-rs5743314-rs5743315 was significantly associated with HBV infection in Saudi Arabian (*Al-Qahtani et al., 2012*). The haplotype GT carriers of rs1879026-rs3775290 had a reduced risk of developing chronic hepatitis B, LC and HCC; in contrast, the haplotype GC was associated with a significantly increased risk of LC (*Huang et al., 2015*).

There is controversy about the association between the rs3775290 polymorphism and HBV. *Goktas et al. (2016)* reported that the HBV DNA levels in patients with the TT genotype were significantly higher than those with the CC and CT genotypes in patients with chronic hepatitis B, suggesting that the TT genotype was a risk factor for chronic hepatitis B infection. This was confirmed in a Tunisian study, which found that the T allele was slightly associated with chronic HBV infection and increased the risk of HBV-associated HCC (*Sghaier et al., 2019*). However, *Huang et al. (2015)* suggested that the TT genotype may be a protective factor for chronic hepatitis B, HBV-related LC and HCC in Chinese. Another study (*Gao et al., 2015*) found that the TT genotype reduced the risk of HBV intrauterine transmission. This contradiction may be due to ethnicity differences in the polymorphism, and should be validated in different population cohorts.

The rs3775290 polymorphism is a non-synonymous mutation located in the fourth exon of the *TLR3* gene on chromosome 4, affecting the interaction between the receptor and the ligand by altering the extracellular domain of TLR3 and impairing the receptor function.

**Table 1** SNPs of the TLR3 gene associated with HBV.

| Polymorphism | Author | Year | population | Sample size | | MAF(%) (controls) | Influence on | References |
|---|---|---|---|---|---|---|---|---|
| | | | | cases | controls | | | |
| rs1879026 (G/T) | Al-Qahtani et al | 2012 | Saudi Arabian | 707 | 600 | 17.20 | Susceptibility to HBV infection | Al-Qahtani et al. (2012) |
| | Huang et al | 2015 | Chinese Han | 437 | 186 | 28.00 | The risk of HBV-related liver diseases | Huang et al. (2015) |
| rs1879026-rs5743313-rs5743314-rs5743315 haplotype | Al-Qahtani et al. | 2012 | Saudi Arabian | 707 | 600 | – | Susceptibility to HBV infection | Al-Qahtani et al. (2012) |
| rs3775290 (C/T) | Goktas et al. | 2016 | Turkish | 116 | 50 | 28.00 | Susceptibility to chronic hepatitis B | Goktas et al. (2016) |
| | Sghaier et al. | 2019 | Tunisian | 274 | 360 | 35.60 | The risk of HBV-related HCC. | Sghaier et al. (2019) |
| | Huang et al. | 2015 | Chinese Han | 437 | 186 | 41.90 | The risk of HBV-relatedliver diseases | Huang et al. (2015) |
| | Gao et al. | 2015 | Chinese | 51 | 348 | 33.30 | Susceptibility to HBV intrauterine transmission | Gao et al. (2015) |
| rs3775291 (C/T) (G/A) | Ye et al. | 2020 | Meta-Analysis | – | – | – | Susceptibility to HBV infection | Ye et al. (2020) |
| | Rong et al. | 2013 | Chinese Han | 452 | 462 | 25.40 | Susceptibility to chronic hepatitis B infection and HBV-related ACLF | Rong et al. (2013) |
| | Geng et al. | 2016 | Meta-Analysis | 3547 | 2797 | – | The risk of HBV-related liver diseases | Geng et al. (2016) |
| | Li et al. | 2013 | Chinese | 466 | 482 | 25.82 | Susceptibility to HBV-related HCC. | Li & Zheng (2013) |
| | Chen et al. | 2017 | Chinese Han male | 688 | 686 | 36.95 | Susceptibility to HBV-related HCC. | Chen et al. (2017) |
| | Fischer et al. | 2018 | Caucasian | 860 | 254 | 24.00 | HBV clearance and susceptibility to HBV infection | Fischer et al. (2018) |
| | Sa et al. | 2015 | Brazilian | 109 | 299 | 33.10 | Susceptibility to HBV infection | Sa et al. (2015) |
| rs5743305 (T/A) | Fischer et al. | 2017 | Caucasian | 494 | 1057 | – | Spontaneous HBsAg clearance | Fischer et al. (2018) |
| | Chen et al. | 2017 | Chinese Han male | 688 | 686 | 30.05 | Susceptibility to HBV infection and HBV-related HCC. | Chen et al. (2017), Li & Zheng (2013), Sa et al. (2015) |

**Notes.**
Abbreviations: MAF, minor allele frequency; HCC, Hepatocellular carcinoma; ACLF, acute-on-chronic liver failure.
Therefore, it is believed that the mutation may passivate TLR3 signaling during the viral infection (*Sghaier et al., 2019*; *Gao et al., 2015*). The reduction of TLR3 activity leads to impaired pathogenic microorganism recognition and inadequate immune response, thus increasing the disease's infectiousness (*Geng et al., 2016*).

A meta-analysis showed that the rs3775291 polymorphism was associated with HBV susceptibility, with the T allele and TT genotype increasing the risk of HBV infection (*Ye et al., 2020*). *Rong et al. (2013)* found that patients with the CT and TT genotypes had 1.4-fold and 2.3-fold higher risk of developing chronic hepatitis B than those with the CC genotypes, respectively. In addition, they also increased the risk of chronic hepatitis B-related acute-on-chronic liver failure (ACLF). A meta-analysis (*Geng et al., 2016*) showed that rs3775291 was associated with a significantly increased risk of HBV-related liver disease. *Li & Zheng (2013)* found that it was associated with increased susceptibility to HBV-related HCC, but *Chen et al. (2017)* showed that the polymorphism may be a protective factor for HBV-related HCC. It reported that the rs3775291 GG genotype was associated with spontaneous HBsAg clearance and spontaneous HBeAg clearance, while the A allele was associated with an increased risk of chronic HBV infection (*Fischer et al., 2018*). In a Brazilian cohort (*Sa et al., 2015*), the rs3775291 polymorphism was not relevant to HBV susceptibility. This discrepancy may be caused by differences in the included study populations and sample size, or more likely, caused by ethnic differences.

The rs3775291 is a missense mutation that results in substituting phenylalanine (Phe) for leucine (Leu) at position 412 of the TLR3 protein. In vitro experiments showed that the function of TLR3 was significantly weakened in the presence of the SNP (*Ranjith-Kumar et al., 2007*). The expression of TLR3 in HCC tissues may have a synergistic effect on apoptosis and inhibit the proliferation of HCC cells and angiogenesis (*Yuan et al., 2015*). Therefore, it is speculated that the decreased function of TLR3 may lead to the up-regulation of vascular remodeling and liver tumor growth. In addition, the minor A allele of rs3775291 was associated with reduced NF-κB activation to half the level of the G allele (*Zhou et al., 2011b*) and reduced type I interferon signaling (*Gorbea et al., 2010*), which may also contribute to reduced cytokine and chemokine production.

*Fischer et al. (2017)* found that the rs5743305 GG genotype was associated with spontaneous HBsAg clearance. However, multiple studies (*Chen et al., 2017*; *Li & Zheng, 2013*; *Sa et al., 2015*) have shown that the polymorphism was not associated with HBV susceptibility and HBV-related HCC.

### *TLR4* gene

The selected studies found four SNPs associated with HBV in *TLR4* gene, including rs4986790, rs4986791, rs11536889 and rs2149356 (Table S2).

The rs4986790 polymorphism is a missense mutation located in the coding region, which causes aspartic acid (Asp) of the 299th amino acid of TLR4 protein to be replaced by glycine (Gly). It is located in the extracellular domain of TLR4, closing to the binding site of TLR4 myeloid differentiation protein-2 (MD-2) (*Ferwerda et al., 2008*). The polymorphism does not affect TLR4 protein expression and subcellular localization. However, it impairs the signal of TLR4 in response to lipopolysaccharide (LPS) (*Long et al., 2014*), which suggests

that the polymorphism may affect TLR4 and MD-2 and LPS and/or the ability of signals to induce interactions.

*Sghaier et al. (2019)* found that the rs4986790 G allele was significantly associated with an increased risk of chronic HBV infection, and patients with the G allele had an approximately 11-fold increased risk of HBV infection. The GG genotype was positively associated with HBV-related HCC. In another study, the G allele was associated with spontaneous serum clearance and serum conversion of HBsAg (*Wu et al., 2012*). The wild AA genotype may be related to the severity of HBV-related liver fibrosis in men, and thus affect the prognosis of chronic HBV infection (*Cussigh et al., 2013*). However, *Pires-Neto Ode et al. (2015)* recently reported that the polymorphism may not be associated with HBV susceptibility in Brazilians. Therefore, the association between the polymorphism and HBV susceptibility needs further study and verification.

*Pires-Neto Ode et al. (2015)* showed that there may be no association in the Brazilian population between the rs4986791 polymorphism and HBV susceptibility.

*Zhou et al., (2011a)* found that the rs11536889 polymorphism was strongly associated with protecting against HBV recurrence after liver transplantation. The CG and GG genotypes carriers had a lower HBV recurrence rate after liver transplantation compared with the CC genotype. In another study (*Zhang et al., 2016*), the CG and GG genotypes were independent risk factors for HCC in patients with chronic hepatitis B cirrhosis. Stratified analysis showed that the rs11536889 G allele and the rs2149356 A allele were independent risk factors for HCC in patients with chronic hepatitis B cirrhosis without a drinking history.

The rs11536889 polymorphism located in the 3'-untranslated region (3'-UTR) of TLR4 mRNA can alter the post-transcriptional regulation of 3'-UTR and the expression of target genes to attenuate the transmembrane signal transduction induced by lipopolysaccharide (*Zhou et al., 2011a*; *Duan et al., 2007*).

### *TLR5* and *TLR7* genes

Four SNPs associated with HBV were found in *TLR5* and *TLR7* genes (Table S3).

*Katrinli et al. (2018)* reported that the rs5744174 polymorphism of the *TLR5* gene was significantly associated with chronic hepatitis B infection in Turkish populations, and the TT genotype may be a protective genotype against HBV infection. It has also been found that the T allele was associated with early spontaneous serum conversion of HBeAg, and the TT genotype was associated with higher IFN-γ secretion (*Wu et al., 2012*). A recent study has suggested that the polymorphism may play a role in developing severe HBV-related liver disease (*Cao et al., 2017*). The mutation of rs5744174 results in the 616th amino acid of the TLR5 protein changing from phenylalanine (Phe) to leucine (Leu). Its influence on HBV susceptibility may be due to protein sequence and structure changes rather than changes in expression levels (*Wu et al., 2012*; *Dhiman et al., 2008*).

Recently, *Zhu et al. (2017)* conducted a study on the SNPs of TLR7, and showed that the rs179009 A allele was associated with a reduced risk of disease progression from chronic hepatitis B to HCC. In contrast, the frequency of the G allele increased significantly as the disease progressed. It is shown that the G allele was a risk factor for the progression

of HBV-related liver disease (chronic hepatitis B, LC and HCC). According to haplotype analysis, the haplotype ACT (rs179009-rs179010-rs2074109) was significantly associated with chronic hepatitis B susceptibility in Chinese males, and the haplotype GCT was a risk factor for the progression of HBV-related liver disease.

### TLR9 gene

In the *TLR9* gene, we found three SNPs associated with HBV (Table S4).

*Chihab et al. (2019)* found that the rs187084 G allele had a protective effect on the progression of HBV infection, and the AA genotype may be significantly associated with the progression of HBV infection to advanced liver disease (AdLD). In addition, it was also significantly associated with the HBV DNA load as the patients with an AA genotype had a higher DNA load than those with the AG genotype. It was also found that the rs5743836 polymorphism was associated with the HBV DNA load and patients with the AA genotype had lower a DNA load than patients with the GG genotype (*Chihab et al., 2019*). Furthermore, *Wu et al. (2012)* reported that the rs5743836 CT genotype was conducive to the early spontaneous seroconversion of HBeAg.

The rs5743836 polymorphism is located in the promoter region of TLR9 gene and is involved in the transcriptional regulation of TLR9 mRNA. Its C allele has been shown to enhance promoter activity and promote TLR9 expression, enhancing thymosinα-1 signaling and inhibiting HBV replication through downstream cytokines (*Lange et al., 2011*).

*He et al. (2015)* showed that the rs352140 polymorphism was associated with the susceptibility of chronic HBV infection, and the relative risk of the CT genotype carriers was lower than that of the CC and TT genotypes. The A allele of the polymorphism (rs352140) was also found to be significantly associated with the susceptibility of HBV intrauterine transmission, and the GA genotype was protective for HBV intrauterine transmission in newborns (*Gao et al., 2015*).

### SNPs of Toll-like receptors associated with HCV
### TLR2 gene

*Neamatallah et al. (2020)* showed that the rs13105517 A allele was associated with the risk of chronic hepatitis C. The rs3804099 C allele was strongly associated with the risk of HCV-related HCC. The haplotype ATAC was significantly associated with chronic HCV infection and HCC development compared with haplotype GCGT (rs1898830-rs1816702-rs13105517-rs3804099).

*Nischalke et al. (2012)* reported that TLR2 -196 to -174 del/ins polymorphism increased the risk of HCC in patients with HCV (genotype 1) infection, and -196 to -174 del affected the HCV viral load. In vitro experiments showed that the impaired upregulation of TLR2 was associated with the -196 to -174 del allele and decreased IL-8 secretion in carriers. Therefore, the authors hypothesized that the reduction of the TLR2 signal may lead to a less effective immune response, resulting in an increased viral load and an increased risk of developing HCC (Table S5).

### *TLR3* gene

In the *TLR3* gene, we summarized eleven major SNPs associated with HCV (Table 2).

Recently, some studies have reported that the rs3775290 T allele was associated with an increased risk of chronic HCV infection. The CT and TT genotypes may be risk factors for chronic HCV infection, while the CC genotype had a protective effect (*Sghaier et al., 2019*; *Mosaad et al., 2019*). It was also found that the C allele was a protective factor for chronic HCV infection in Egyptians and Tunisian cohorts and were positively associated with enhanced viral clearance during antiviral therapy (*El-Bendary et al., 2018*; *Sghaier et al., 2017*). However, different results were obtained in Egyptians. *Hamdy et al. (2018)* found that the CC genotype was associated with chronic HCV infection, but another study reported that it was not associated with HCV susceptibility (*Pandey et al., 2011*). Study populations that differ according to sample sizes, ethnic differences, grouping differences and other factors may cause variations in the results. The results should be verified in different ethnic groups and large sample size cohorts.

The rs3775290 is a non-synonymous SNP, resulting in the substitution of leucine (Leu) for phenylalanine (Phe) at the 459th position of the TLR3 protein (*Pandey et al., 2011*). This is associated with reduced activation of the interferon regulatory factor 3(IRF-3), which leads to a decrease in the production of type I IFN (*Zhang et al., 2007*). This phenomenon is thought to promote the proliferation of the virus and the response to antiviral therapy is poor.

Research on the rs3775291 polymorphism is still controversial at present. Some studies have shown that the CC genotype can increase the risk of HCV infection because the T allele of the rs3775291 polymorphism can reduce the risk of HCV infection, and the CC genotype was associated with the severe recurrence of HCV infection after transplantation (*Geng et al., 2016*; *Citores et al., 2016*). *Citores et al. (2011)* reported that the polymorphism also played an important role in acute rejection in liver transplantation patients with HCV-related cirrhosis. Patients with the TT genotype were associated with a lower rate of acute rejection compared with those with the CC genotype. However, another study showed a significant association between the CT genotype and liver transplantation failure and mortality (*Lee, Brown & Razonable, 2013*). A study in HCV/HIV co-infected patients also found that the rs3775291 polymorphism was associated with HCV treatment response in patients (mainly those infected with HCV genotypes 2 and 3). The minor A allele was associated with a decreased odds of achieving a virologic response to HCV therapy, whereas the G allele was associated with a significantly higher proportion of successful virologic response (where G was the favorable allele) (*Jiménez-Sousa et al., 2015*). However, *El-Bendary et al. (2018)* found that the C allele was a protective factor for HCV infection in Egyptians, whereas *Sa et al. (2015)* did not observe this association in the Brazilian cohort.

*Jiménez-Sousa et al. (2015)* reported that the minor A allele of rs13126816 was associated with decreased odds of a virologic response to HCV therapy in HCV/HIV co-infection patients, while the major G allele was a protective factor. In addition, haplotype analysis revealed that haplotype GG (rs3775291- rs13126816) was associated with a sustained virologic response (no detectable HCV viral load (<10 IU/mL) six months after treatment cessation) in all co-infected patients and those infected with HCV genotypes 2 and 3.

**Table 2  SNPs of the TLR3 gene associated with HCV.**

| Polymorphism | Author | Year | population | Sample size | | MAF(%) (controls) | Influence on | References |
|---|---|---|---|---|---|---|---|---|
| | | | | cases | controls | | | |
| rs3775290 (C/T) | Sghaier et al. | 2019 | Tunisian | 274 | 360 | 35.60 | Susceptibility to HCV infection | Sghaier et al. (2019), Mosaad et al. (2019) |
| | El-Bendary et al. | 2018 | Egyptian | 1908 | 1460 | – | Susceptibility to HCV infection; the outcome of chronic hepatitis C treatment | El-Bendary et al. (2018), Sghaier et al. (2017) |
| | Hamdy et al. | 2018 | Egyptian | 281 | 265 | 23.40 | Susceptibility to HCV infection | Hamdy et al. (2018) |
| | Zayed et al. | 2017 | Egyptian | 100 | 100 | 25.50 | Susceptibility to HCV infection | Zayed et al. (2017) |
| rs3775291 (C/T) | Geng et al. | 2016 | Meta-Analysis | 3547 | 2797 | – | Susceptibility to HCV infection; HCV recurrence infection after transplantation | Geng et al. (2016), Citores et al. (2016) |
| | Citores et al. | 2011 | Spanish | 37 | 63 | 42.86 | Acute rejection of liver transplantation for HCV-related cirrhosis | Citores et al. (2011) |
| | Lee et al. | 2013 | Caucasian (mostly) | 153 | 458 | 25.9 | HCV-related liver transplant | Lee, Brown & Razonable (2013) |
| | Jiménez-Sousa et al. | 2015 | Spanish | 321 | – | – | Virological response to HCV treatment | Jiménez-Sousa et al. (2015) |
| | El-Bendary et al. | 2018 | Tunisian | 1908 | 1460 | – | Susceptibility to HCV infection | El-Bendary et al. (2018) |
| | Sa et al. | 2015 | Brazilian | 109 | 299 | 33.10 | Susceptibility to HCV infection | Sa et al. (2015) |
| rs13126816 (G/A) | Qian et al. | 2013 | Whit, Black and Hispanic | 32 | 37 | – | HCV spontaneous clearance | Qian et al. (2013) |
| rs78726532 (A/G) | Al-Anazi et al. | 2017 | Saudi Arabian | 563 | 599 | 12.02 | Susceptibility to HCV infection | Al-Anazi et al. (2017) |
| rs5743313 (C/T) | Al-Anazi et al. | 2017 | Saudi Arabian | 563 | 599 | 27.88 | The progression of HCV-related liver diseases (LC and HCC) | Al-Anazi et al. (2017) |

Xu et al. (2022), *PeerJ*, DOI 10.7717/peerj.13335

**Table 2** (*continued*)

| Polymorphism | Author | Year | population | Sample size | | MAF(%) (controls) | Influence on | References |
|---|---|---|---|---|---|---|---|---|
| | | | | cases | controls | | | |
| rs5743314 (G/C) | | | | | | 28.13 | | |
| rs111611328 (G/C) | | | | | | 1.00 | | |
| rs5743312 (C/T) | El-Bendary et al. | 2018 | Egyptian | 1908 | 1460 | – | Susceptibility to HCV infection | *El-Bendary et al. (2018)* |
| −705 (A/G) | Medhi | 2011 | Indian | 180 | 180 | 4.00 | Susceptibility to HCV infection | *Medhi et al. (2011)* |
| rs3775296 (C/A) | Medhi | 2011 | Indian | 180 | 180 | 4.00 | Susceptibility to HCV infection | *Medhi et al. (2011)*, *Zayed et al. (2017)* |
| rs5743305 (T/A) | Sa et al. | 2015 | Brazilian | 109 | 299 | 29.25 | Susceptibility to HCV infection | *Sa et al. (2015)* |

**Notes.**

Abbreviations: MAF, minor allele frequency; LC, liver cirrhosis; HCC, Hepatocellular carcinoma.

The rs13126816 polymorphism had an effect on the expression of TLR3 in macrophages, and the GG genotype was conducive to the spontaneous clearance of HCV (*Qian et al., 2013*). *Al-Anazi et al. (2017)* also found that the rs78726532 GG genotype was protective against HCV infection in Saudi Arabians. The rs5743313, rs5743314 (GC genotype) and rs111611328 polymorphisms were closely associated with the progression of hepatitis C-related liver diseases (LC and HCC). Furthermore, some studies have reported that the rs5743312 C allele was a protective factor for HCV infection in Tunisians (*El-Bendary et al., 2018*), the G allele of the -705 (A/G) polymorphism in the promoter region increased the risk of HCV infection (*Medhi et al., 2011*), the rs3775296 polymorphism had no association with HCV infection in Egyptians and Indian (*Medhi et al., 2011*; *Zayed et al., 2017*), and the rs5743305 polymorphism was not associated with HCV susceptibility in Brazilians (*Sa et al., 2015*). The association between the SNPs of *TLR3* and HCV infection in different ethnicities, different nationalities and larger sample groups required further study.

### *TLR4* gene

In the *TLR4* gene, we found eleven SNPs associated with HCV (Table 3).

*Peric et al. (2015)* reported that the rs4986790 polymorphism was associated with the HCV load. Some studies have also shown that the polymorphism was associated with the risk of HCV infection. The A allele and the AA and AG genotypes had a protective effect on chronic HCV infection, while patients with the G allele were almost 6-fold more likely to be infected with HCV than other patients (*Sghaier et al., 2019*; *Neamatallah et al., 2020*; *Sghaier et al., 2017*; *Chaiwiang & Poyomtip, 2019*). In contrast, the G allele was protective in a Saudi Arabian cohort (*Al-Qahtani et al., 2014*). *Iqbal et al. (2017)* found that the polymorphism was associated with an increased risk of HCV infection in Pakistan, but the opposite result was obtained in a Brazilian cohort (*Pires-Neto Ode et al., 2015*). There is still controversy in the studies of the association between rs4986790 polymorphism and hepatitis C related liver diseases. Some studies had shown that the G allele and the GG genotype were associated with an increased risk of hepatitis C-related diseases (*Sghaier et al., 2019*). *Dhillon et al. (2010)* reported that the rs4986790 polymorphism was significantly associated with poor liver graft survival in HCV-infected recipients, with the G allele leading to worse graft survival. However, *Al-Qahtani et al. (2014)* showed no association. The contradiction may be related to ethnic differences, sample size differences, different grouping methods and others, and should be verified in additional cohorts.

The non-structural 5A protein of HCV inhibits LPS-induced hepatocyte apoptosis by down-regulating the expression of TLR4, thereby participating in the pathogenesis of an HCV infection or evading immune surveillance (*Tamura et al., 2011*). The rs4986790 polymorphism is located at the ligand-binding site of TLR4, but this mutation does not affect the binding of LPS (*Ohto et al., 2012*). According to the research, it may lead to the inefficient recruitment of MyD88 and TRIF by TLR4 and impair the TLR4 signaling pathway, thus playing a role in HCV pathogenesis (*Figueroa et al., 2012*). Therefore, it may affect other properties of TLR4 between host and virus interactions.

In the studies of the rs4986791 polymorphism, some have found that it was associated with HCV load (*Peric et al., 2015*). Other studies have shown that the T allele had a
**Table 3  SNPs of the TLR4 gene associated with HCV.**

| Polymorphism | Author | Year | population | Sample size | | MAF(%) (controls) | Influence on | References |
|---|---|---|---|---|---|---|---|---|
| | | | | cases | controls | | | |
| rs4986790 (A/G) | Perić et al. | 2015 | Croatian | 60 | 40 | 17.50 | HCV load | *Peric et al. (2015)* |
| | Sghaier et al. | 2019 | Tunisian | 274 | 360 | 19.02 | Susceptibility to HCV infection | *Sghaier et al. (2019), Neamatallah et al. (2020), Sghaier et al. (2017), Chaiwiang & Poyomtip (2019)* |
| | Al-Qahtani et al. | 2014 | Saudi Arabian | 450 | 600 | 10.60 | Susceptibility to HCV infection | *Al-Qahtani et al. (2014)* |
| | Iqbal et al. | 2017 | Pakistani | 400 | 100 | 14.00 | Susceptibility to HCV infection | *Iqbal et al. (2017)* |
| | Pires-Neto et al. | 2015 | Brazilian | 121 | 299 | 4.01 | Susceptibility to HCV infection | *Pires-Neto Ode et al. (2015)* |
| | Sghaier et al. | 2019 | Tunisian | 274 | 360 | 19.02 | The risk of HCV-related liver diseases | *Sghaier et al. (2019)* |
| | Dhillon et al. | 2010 | Caucasian (71%) | 430 | – | – | Liver graft survival of HCV-infected recipients | *Dhillon et al. (2010)* |
| rs4986791 (C/T) | Perić et al. | 2015 | Croatian | 60 | 40 | 17.50 | HCV load | *Peric et al. (2015)* |
| | Al-Qahtani et al. | 2014 | Saudi Arabian | 450 | 600 | 11.00 | Susceptibility to HCV infection | *Al-Qahtani et al. (2014)* |
| | Pires-Neto et al. | 2015 | Brazilian | 121 | 299 | 4.01 | Susceptibility to HCV infection | *Pires-Neto Ode et al. (2015), Neamatallah et al. (2020), Chaiwiang & Poyomtip (2019)* |
| rs2149356 (G/T) | Chaiwi-ang et al. | 2019 | Meta-Analysis | 391 | 430 | 31.86 | Susceptibility to HCV infection | *Chaiwiang & Poyomtip (2019)* |
| | Agunde-z et al. | 2012 | Spaniard | 308 | 390 | 52.30 | The risk of HCV-related HCC | *Agundez et al. (2012), Sadik et al. (2015)* |

**Table 3** (*continued*)

| Polymorphism | Author | Year | population | Sample size | | MAF(%) (controls) | Influence on | References |
|---|---|---|---|---|---|---|---|---|
| | | | | cases | controls | | | |
| rs10759930 (T/C) | Shi et al. | 2011 | Chinese Han | 216 | 228 | 43.42 | The risk of HCV-related HCC | *Minmin et al. (2011)* |
| rs2737190 (A/G) | | | | | | 42.11 | | |
| rs10116253 (T/C) | | | | | | 42.11 | | |
| rs1927914 (T/C) | | | | | | 41.67 | | |
| rs12377632 (C/T) | | | | | | 41.23 | | |
| rs1927911 (C/T) | | | | | | 42.11 | | |
| rs10116253 (T/C) | Neamat-allah et al. | 2020 | Egyptian | 1680 | 1615 | 15.00 | Susceptibility to HCV infection and the risk of HCV -related HCC | *Neamatallah et al. (2020)* |
| rs5030728 (G/A) | | | | | | 8.00 | | |

**Notes.**

Abbreviations: MAF, minor allele frequency; HCC, Hepatocellular carcinoma.

protective effect on HCV infection, but it was not associated with HCV-related liver diseases (*Al-Qahtani et al., 2014*) and some studies have reported that it was not associated with HCV susceptibility (*Pires-Neto Ode et al., 2015*; *Neamatallah et al., 2020*; *Chaiwiang & Poyomtip, 2019*).

A meta-analysis suggested that the rs2149356 TG genotype may be a protective factor for HCV infection (*Chaiwiang & Poyomtip, 2019*). The T allele was associated with reduced HCC risk and may slow the clinical progression of chronic liver disease caused by HCV, which was validated in an Egyptian cohort (*Agundez et al., 2012*; *Sadik et al., 2015*). According to in silicon analysis, the rs2149356 G allele can produce miRNA, and the primary functional annotation of the target indicated that the enrichment of miRNA was highest in the autophagy pathway (*Chaiwiang & Poyomtip, 2019*). It is worth noting that HCV can inhibit innate immunity and cell death in the host by inducing autophagy (*Taguwa et al., 2011*; *Chan & Ou, 2017*). Therefore, it is speculated that the rs2149356 TG genotype may alter the autophagy pathway and lead to the limitation of HCV infection.

*Minmin et al. (2011)* found that the individuals carrying the heterozygous genotypes of the rs10759930 (CT), rs2737190 (AG), rs10116253 (CT), rs1927914 (CT), rs12377632 (CT) and rs1927911 (CT) in *TLR4* had a significantly reduced risk of HCC. A recent study reported that the rs10116253 C allele and the rs5030728 A allele were strongly associated with chronic hepatitis C and the development of HCV-related HCC. The former was also associated with HCV susceptibility. In haplotype analysis, the haplotype CAGT of *TLR4* gene was significantly associated with chronic progression of HCV and the progression of HCC compared with TGAC (rs10116253-rs5030728-rs4986790-rs4986791) (*Neamatallah et al., 2020*).

### *TLR7* gene

The main SNPs in the *TLR7* gene associated with HCV are summarized in Table S6.

The influence of the rs3853839 polymorphism on HCV susceptibility was related to gender, and its C allele was a protective factor for female HCV infection. In the detection of TLR7 mRNA and cytokines, it was found that the CC genotype had significantly lower mRNA levels, decreased IFN-α secretion and increased IL-6 production compared with the G allele carriers (*El-Bendary et al., 2018*; *Yue et al., 2014a*).

Patients with the rs179008 A allele were more likely to spontaneously clear the virus, while the T allele increased the risk of progression to AdLD (*El-Bendary et al., 2018*), which was demonstrated in an Egyptian cohort (*El-Bendary et al., 2018*). However, the study found no significant association between the polymorphism and HCV-related inflammatory activity or fibrosis progression. It did not affect HCV load but decreased the expression of IL-29 /IFN-λ (*Askar, Ramadori & Mihm, 2010*).

Research on the rs179009 polymorphism are controversial. *Yue et al. (2014b)* found that the GG genotype was associated with an increased risk of HCV infection in Chinese women. This was consistent with *Fakhir et al. (2018)*, who found that the G allele increased the risk of disease progression, while the A allele increased the spontaneous virus clearance in female patients by a factor of two. However, it has been reported that the AA genotype

may be a susceptibility factor for chronic HCV infection in Chinese Han females, while the AG genotype was a protective factor (*Wei et al., 2014*).

In addition, *Xue et al. (2015)* showed that the rs179016 C allele hurt the spontaneous clearance of HCV in male patients, who were more likely to develop persistent infection. The rs1634323 G allele had a protective effect on persistent HCV infection in female patients. In haplotype analysis, the haplotype CCA (rs179016-rs5743733-rs1634323) increased the risk of HCV infection in women, while the GGA increased the risk of HCV infection in men. The haplotype GCG (rs179009-rs179010-rs179012) was found to be associated with increased HCV susceptibility (*Yue et al., 2014b*).

### *TLR8* and *TLR9* genes

In the two genes, the main HCV-related SNPs are summarized in Table S7.

A study in an Egyptian cohort showed that the C allele of the TLR8 rs3764879 polymorphism was a protective factor for HCV infection in men, and the TLR8 rs3764880 A allele was a risk factor for HCV infection in both men and women (*El-Bendary et al., 2018*). These results supported those of *Fakhir et al. (2018)* who found that the rs3764880 G allele was associated with spontaneous HCV clearance. In men, the rs3764879 C and rs3764880 A alleles were significantly associated with advanced liver disease (AdLD). In addition, in haplotype analysis, the haplotype GG (rs3764879-rs3764880) was protective against HCV infection (*Wang et al., 2014*). The rs3764880 polymorphism inactivated the start codon from ATG to GTG, resulting in a shorter TLR8 mRNA, which may lead to its faster decay or possibly affect protein function, thereby affecting the immune process of HCV infection (*Oh et al., 2008*).

A previous study found that the haplotype AG (*TLR7* rs179009-*TLR8* rs3764879) had a protective effect on chronic HCV infection, leading to reduced IFN-α secretion and increased production of pro-inflammatory cytokines. They also observed that the two polymorphisms altered mRNA expression levels. This may explain the susceptibility of patients carrying these polymorphisms to progress to AdLD (*Wang et al., 2011*). *Fernández-Rodríguez et al. (2015)* showed that the T allele of rs1013151 and rs5744069 polymorphisms were associated with non-progression of liver fibrosis in HIV/HCV co-infected patients, especially in men and HCV genotype 1 infected patients.

Recent studies of the *TLR9* gene have shown that the rs5743836 polymorphism was not associated with HCV infection susceptibility in Egyptians and Pakistanis (*Hamdy et al., 2018*; *Zayed et al., 2017*; *Aslam et al., 2018*). The rs352140 polymorphism was not related to Egyptian exposure to hepatitis C (*Hamdy et al., 2018*). However, *Valverde-Villegas et al. (2017)* reported that the rs352140 AA genotype was associated with the susceptibility of Brazilians to HIV/HCV co-infection. The polymorphism may affect liver pathophysiology and cirrhosis following a HCV genotype 4 infection and its A allele is a poor prognostic factor for liver fibrosis and cirrhosis (*Youssef & Hamdy, 2017*). It is worth noting that *Clausen et al. (2014)* reported that carriers of the rs352140 A allele were associated with spontaneous HCV resolution in a Caucasian cohort. Still, the association did not reach statistical significance in the validated cohort. It was also found that the rs187084 polymorphism was associated with significant differences in HCV clearance rates

in men and women in a German cohort. In women, the C allele may be associated with a 1.9−2.2-fold increase in spontaneous HCV clearance compared with carriers of the TT genotype, suggesting that it may facilitate spontaneous viral clearance (*Fischer et al., 2017*).

## Conclusion and Perspectives

An increasing number of studies have shown that TLRs polymorphisms play an important role in HBV and HCV infection. Understanding these gene expression profiles and single nucleotide polymorphisms may be important for elucidating the pathological mechanisms of HBV and HCV susceptibility and related liver disease progression.

Presently, the increasing research on the biological effects of genetic polymorphisms have uncovered that genetic polymorphisms have a very important effect on viral infectious diseases. SNPs are important for considering individual susceptibility to viral infectious diseases, drug sensitivity and adverse reactions, the development of personalized therapeutic strategies, and the identification of new therapeutic targets. First, genetic polymorphisms associated with viral susceptibility or its related diseases progression are critical for identifying high-risk populations. Only in this context can we develop effective prevention strategies to combat, reduce and even avoid new infections (*Ellwanger et al. , 2018*). A study has shown that the TLR2 rs3804100 polymorphism was significantly associated with the status of serum anti-HBV response induced by vaccines (*Chen et al., 2011*). By detecting these related polymorphisms, we can effectively improve the immune efficiency of vaccines. At the same time, for high-risk populations with low immune efficiency, other effective preventive measures can be used to effectively reduce the incidence of viral infectious diseases. Second, genetic polymorphisms may help predict the clinical course of diseases. It has been reported that among the SNPs in TLR3, the rs13126816 GG genotype was favorable for spontaneous HCV clearance (*Qian et al., 2013*), and the rs3775291 CT genotype was significantly associated with liver transplantation failure and mortality (*Lee, Brown & Razonable, 2013*). Based on these studies, clinicians may be able to provide patients with appropriate measures early in the disease to improve adverse outcomes. However, whether TLRs SNPs can predict the clinical course of the disease still needs to be explored in future studies. Third, we can develop more effective therapies based on the biological effects of genetic polymorphisms. *Jiménez-Sousa et al. (2015)* found that the minor A allele of rs13126816 decreased the odds of a virologic response to HCV therapy in HCV/HIV co-infected patients. This suggests that we should pay attention to the influence of genetic polymorphisms on the therapeutic response of viral infectious diseases in future researches. A better understanding of this will facilitate the development of more powerful therapeutic plans to compensate for the adverse effects of genetic polymorphisms and achieve the individualized treatment of viral infectious diseases.

Many studies have been included in this review on the association between SNPs of TLR2, TLR3 and TLR4 and HBV and HCV infection. A study has shown that HBV induced the inflammatory response of monocytes through the TLR2/MyD88/NF-κB signaling pathway (*Song et al., 2019*). The expression of TLR2 and TLR4 was significantly down-regulated in peripheral blood mononuclear cells of HBV-infected patients, resulting in impaired cytokine production (*Chen et al., 2008*). However, in the monocytes of patients

with chronic hepatitis C, the expression of TLR2 and TLR4 were up-regulated, and TLR2-mediated pro-inflammatory cytokines, such as TNF-α, are increased (*Riordan et al., 2006*). However, the SNPs of TLRs can affect their role in HBV and HCV infection. In vitro experiments showed that the −196 to −174 del allele resulted in impaired upregulation of TLR2 expression and decreased TLR2 signaling lead to poor immune response and increased HCV load (*Nischalke et al., 2012*). TLR4 rs4986790 polymorphism is a missense mutation located in the coding region in the ligand binding site of TLR4 (*Ferwerda et al., 2008*). This may lead to the inefficient recruitment of MyD88 and TRIF by TLR4 and impaired TLR4 signaling pathway (*Figueroa et al., 2012*), thus playing a role in the pathogenesis of HBV and HCV. In addition, there are many studies on the rs3775290 and rs3775291 polymorphisms of TLR3. The rs3775290 polymorphism is located in the fourth exon, which affects the receptor–ligand interaction by changing the TLR3 extracellular domain and damaging the receptor function, thereby blunting TLR3 signaling during viral infection (*Sghaier et al., 2019*; *Gao et al., 2015*). This was associated with decreased activation of IRF-3 and decreased production of type I IFN (*Zhang et al., 2007*), increasing viral infectivity. In vitro experiments also showed that the function of TLR3 was significantly weakened in the presence of rs3775291 polymorphism (*Ranjith-Kumar et al., 2007*). However, the results of studies on the role of these polymorphisms in HBV and HCV infection have been inconsistent. Therefore, we still need to focus on their role in different populations and signaling pathways in the future. More effective measures are needed to solve a series of problems encountered in the prevention, infection and treatment of HVB and HCV to reduce the infection rate and improve the treatment response.

To date, the role of TLRs single nucleotide polymorphisms in HBV and HCV infection are still controversial. There are few studies on how TLRs' SNPs affect susceptibility to HBV and HCV, so its mechanism of action is still unclear and requires further study. Functional studies of the mechanism of SNPs in disease are needed to better understand the effects of TLRs SNPs on HBV and HCV susceptibility and related liver diseases, and to further clarify its mechanism of action. These results will play an important guiding role in disease prevention, treatment, outcome and prognosis monitoring.

Asia and Africa are the most concentrated areas in countries with high prevalence of chronic hepatitis B (*Trépo, Chan & Lok, 2014*), especially in China (*Liu & Fan, 2007*). Therefore, when reviewing the literature on the effect of TLRs-related polymorphisms on HBV and HCV susceptibility and related diseases, it is not surprising that a large number of studies involving Asian populations were included. However, it leads to discrepancies in the amount of data reported in Asian and non-Asian populations. Therefore, the role of genetic polymorphisms mentioned in the review may not be applicable to other populations. It is necessary to explore the effects of these SNPs on HBV and HBV susceptibility in other populations, especially in non-Asian populations.

It is well known that the same SNP may have different or identical effects in various populations. Different research cohorts have different genetic backgrounds, and SNPs' polymorphism can be altered due to ethnicity, regionalism, and gender differences. Therefore, future research should pay attention to the following points: first, homogenous populations (same ethnicity, same nationality, same region and same gender, etc.) should

be included in the study cohort and studies should be conducted in different homogeneous populations to emphasize the need for validation in different populations. This is conducive to understanding the genetic background of TLRs in a specific population and analyzing the impact of SNPs on the disease. Second, large cohorts should be selected for research to reduce bias. Third, the same SNP polymorphism needs to be studied and validated in similar and different populations to determine its effect (beneficial or harmful) on disease in diverse populations. Fourth, the clinical application of SNPs is still limited because in the human body, multiple genes often interact to influence diseases. Therefore, we should attach importance to multi-locus combined analysis in future studies, namely haplotype analysis, which is more conducive to clinical application.

Future research may allow us to construct a risk prediction model for HBV and HCV infection. This will improve our ability to predict the risk of HBV and HCV infection or the threat of related liver disease progression through multiple TLR single nucleotide polymorphisms, thus playing a role in the prevention, monitoring and even treatment guidance for HBV and HCV infection.

### Funding

This study was supported by the Lanzhou Science and technology project (grant no.2021-1-104). The funders had no role in study design, data collection and analysis, decision to publish, or preparation of the manuscript.

### Grant Disclosures

The following grant information was disclosed by the authors:
The Lanzhou Science and technology project: no. 2021-1-104.

### Competing Interests

The authors declare there are no competing interests.

### Author Contributions

- Yaxin Xu and Wentao Xue conceived and designed the experiments, performed the experiments, prepared figures and/or tables, authored or reviewed drafts of the paper, and approved the final draft.
- Hongwei Gao conceived and designed the experiments, performed the experiments, authored or reviewed drafts of the paper, and approved the final draft.
- Jiabo Cui and Lingzhi Zhao analyzed the data, prepared figures and/or tables, and approved the final draft.
- Chongge You conceived and designed the experiments, authored or reviewed drafts of the paper, and approved the final draft.

### Data Availability

This article is a literature review.

## Supplemental Information

Supplemental information for this article can be found online at http://dx.doi.org/10.7717/peerj.13335#supplemental-information.

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
