# Peer review of "Association of toll-like receptors single nucleotide polymorphisms with HBV and HCV infection: research status"

_PeerJ, doi:10.7717/peerj.13335_

## Round 0.1 · original submission · Major Revisions

All three reviewers raised very significant concerns and found numerous serious flaws. Typically, this would mean rejection. However, I decided to give you an opportunity to revise your manuscript in line with all the comments of the three reviewers. Please also note that all reviewers indicated that your manuscript has numerous linguistic issues and requires extensive editorial work. Therefore I am strongly encouraging you to use a professional editor.

Reviewer 1 ·

Basic reporting

This is a review manuscript aiming to summarize data from TLR gene polymorphisms and HBV and HCV infection. In the Abstract, the authors start mentioning the high incidence of such infections in China, but the review apparently aims to cover studies made in diverse populations (although a clear bias to studies performed in China can be seen in the manuscript).

The Introduction brings general information about TLR and the involvement of these receptors in different pathways and diseases. In fact, sentences such as “The inflammatory signaling pathways mediated by TLRs are associated with a series of liver diseases, ...” are quite shallow since several pathways are involved in susceptibility or development of such diseases, therefore an actual link between these receptors and diseases should be made explicit.

Considering that this is a review, the methodology is correct, although a quick PubMed search revealed several studies not cited.
Only as an example, considering TLR2 and HBV infection, at least two other studies (not mentioned in Table 1) could be found:
Kim et al. Hepatogastroenterology. 2010 Nov-Dec;57(104):1351-5.
Chen et al. Vaccine. 2011 Jan 17;29(4):706-11. doi: 10.1016/j.vaccine.2010.11.023.

Extensive review of the English-language usage is needed.

Experimental design

Considering that this is a review, the methodology is correct, although a quick PubMed search revealed several studies not cited (see examples above).

Validity of the findings

The conclusion is somehow confusing and shallow. See these two contradictory statements in the same Conclusion paragraph: “More and more studies have shown that TLRs polymorphisms play an important role in HBV/HCV infection.” followed by “To date, few studies have explored the effect of TLRs single nucleotide polymorphisms on HBV/HCV susceptibility,...”

The discussion about “homogeneous individuals” should be revised, approaching “populations” instead of individuals, and highlighting also the need of inclusion of diverse populations in order to understand the potential involvement of TLRs in HBC/HCV infection.

Finally, there is some redundancy between the text and Tables. Therefore, I would suggest a reorganization of the manuscript, with data presented in the text and Tables just as a summary of data.

Additional comments

There are several incomplete sentences. For example, line 318 states “The main related SNPs in TLR7 gene summarized in Table 9.” Although Table 9 is not summarizing important TLR7 SNPs in general but solely those evaluated in the context of HCV infection.

Reviewer 2 ·

Basic reporting

- The expression “HBV/HCV infection” in the title and other parts of the text suggests that the authors will address the co-infection by HBC and HCV in the article, which is not the case. This expression should be replaced by “HBV and HCV infections” (or similar). I suggest using the expression “HBV/HCV infection” only to refer to co-infection.

- I understand that HBV and HCV infections are important problems in China. However, the article is not only focused on China, as the abstract suggests. This should be fixed to make the article more interesting to a wider readership.

- Genetic polymorphisms are just a few of the multiple determinants of susceptibility and clinical progression of viral infections. The effects of polymorphisms on these conditions must be interpreted in the context of environmental factors and other biological characteristics of the human host, in addition to viral factors. This information should be mentioned/better discussed in the introduction section. Some suggestions of references: Ellwanger et al. Infect Genet Evol. 2018; 66:376-391. doi: 10.1016/j.meegid.2017.08.011; and Ellwanger et al. Front Immunol. 2018; 9:1316. doi: 10.3389/fimmu.2018.01316

Experimental design

- This is a review article. The “Survey methodology” is satisfactory.

Validity of the findings

- The main body of the review is adequate. However, I suggest the authors point out that there is a discrepancy in the amount of information regarding Asian and non-Asian populations, with a greater amount of data for Asian populations. Further discussion of this discrepancy would make the review more relevant.

- The overall result of the review is very complete and interesting. However, the authors should also better explain to the reader why understanding the effects of genetic polymorphisms on infectious diseases is important. Some guidance: (I) it helps to develop better therapies based on the biological effect of genetic polymorphism (e.g., knowing the effect of CCR5delta32 was critical for the development of anti-HIV therapies; for this discussion/example, see: Ellwanger et al. Virus Res. 2020; 286:198040. doi: 10.1016/j.virusres.2020.198040); (II) it allows the identification of populations at higher risk for infection and that need more attention and prevention strategies; (III) genetic polymorphisms may help predict the clinical course of disease (for example, see: Lassner et al. J Transl Med. 2018; 16(1):249. doi: 10.1186/s12967-018-1610-8); among others. Creating a short topic covering these points can be interesting (place it before the conclusion section).

Additional comments

- The authors need to remind you that the name of genes must be written in italics. This should be reviewed and corrected throughout the article.

- The manuscript must be submitted to a professional English review. This point is critical for the improvement of the manuscript.

- Figure 1: authors must give credit for the platform used to create the figure (for this, use figure legend or acknowledgment section).

Reviewer 3 ·

Basic reporting

Manuscript ID: 67425
Name: “Association of Toll-like receptors single nucleotide polymorphism with HBV/HCV infection: research status”

In this manuscript (ms), Xu et al. reviewed the impact of genetic variants in the Toll-like receptor (TLR) genes in HBV and HCV infections. The review was divided into two major sections, HBV, and HCV. Overall, I found the ms difficult to follow at times as important information/background is missing or absent. Also, the lack of standardization and misuse of concepts are concerning. Therefore, the ms needs plenty of work to reach PeerJ readers, and my recommendation is to reject it. Please, see below my report.

Experimental design

1. The Introduction should be improved. Provide additional information regarding how differential expression of TLR may impact overall viral infections, disease progression, treatment, and downstream gene expression. In this context, there is no mention of downstream inflammatory response in Figure 1.
2. L67-70. It should be clearly defined the aims of the review and the topics covered. Also, considering that SNPs are being reviewed it is fundamental to define how or which nomenclature will be used (e.g., position, amino acid change, according to the original study, reference sequence).
3. Survey methodology. The search method needs to be more detailed. Inform the timeframe of your search, from the beginning (?) to the end (June 2021). How many studies were retrieved from in each database? Have the authors considered using MeSH terms?. If not, this might limit the range of search (PMID: 32912361). Please, include a flow diagram informing the total number of studies in each step (data gathering, exclusion, etc.). Lastly, what do the authors mean with “The role of SNPs is not described in detail” as an exclusion criterion? Although it is not a systematic review, one may benefit to follow the PRISMA guidelines (http://prisma-statement.org/).
4. Section 3, 4, and 5. The lack of standardization (SNPs nomenclature) makes the overall comprehension of the review difficult. Also, when discussing the role of an SNP make sure to indicate the allele associated with the risk or protection. Also, there is no mention of how these SNPs contribute to the pathogenesis (regulatory SNP, coding region, etc.).
5. Avoid new definitions. For example, chronic hepatitis B infection is traditionally defined as chronic HBV instead of CHB.
6. Concept misuse. Mutation and polymorphism are not interchangeable. L126, what author mean by racial differences? I would strongly avoid this discussion and use ethnicity/population instead.
7. Tables. Revise the concepts of correlation and association. It is not clear the meaning of “the genotype may be a protective genotype”. Was it associated or not? The sample size column is uninformative. Break it into cases and controls. I suggest including the minor allele frequency of SNPs observed in the control group. Lastly, standardize SNP nomenclature and define all acronyms used.

Validity of the findings

8. Conclusion. I would like to see a more detailed conclusion/discussion of potential TLR’ SNPs involved in HBV, HCV, or both. What is the impact of downstream signaling? Also, how functional studies may push forward the field?

Additional comments

Minor commentaries
1. The ms would benefit from a language editing service.
2. Make sure you list the corresponding author in the main text.
3. L17, HBV, and HCV are major public health burdens worldwide. Unless you are reviewing the role of TLR in a specific population (i.e., China) it is wise to introduce it.
4. Review in-text referencing of literature. For example, L39: replace “. [1-3]” with “[1-3].”.
5. Define acronyms when first cited in the text. See, L54.

---

## Round 0.2 · Major Revisions

Please address the remaining concerns of the reviewers and revise the manuscript accordingly.

Reviewer 1 ·

Basic reporting

Although I acknowledge the efforts of the authors in order to answer all questions raised previously by this reviewer, and the fact that this is an actually better version of the manuscript, still some work should be done.

An extensive review of the English-language usage is needed. There are several typos (example line 218 “mete analysis”) and grammar errors (example: “The natural history of chronic hepatitis B and C are strongly influenced...”), but the most important is to review the manuscript to avoid several confusing statements.

For example, talking about TLRs, author’s claimed that “They recognize different molecular motifs, so called pattern recognition receptors (PRRs).” In fact “They recognize different molecular motifs, called PAMPS and DAMPS” . PAMPs and DAMPs are the viral or bacterial, the external, molecules, although TLRs are PRRs expressed by the host (human) cells. Please revise the text accordingly.

Another example: “For example, the TLR2 rs4696480 polymorphism is located in the promoter region, which may partially inhibit pro-inflammatory cytokines by reducing TLR2 transcription, suggesting that it may have the potential to reduce inflammatory damage in chronic HBV infection [42].” This sentence suggests that this specific SNP is capable to directly inhibit pro-inflammatory cytokines, although the reference cited is an association study. The authors are extrapolating the results from reference 42 . Please revise.

In the sentence “Furthermore, the polymorphism was significantly associated with the status of vaccine-induced serum anti-HBV response, and the TT genotype was a protective factor for anti-non-response.”, please define “anti-non-response”. In the cited reference (number 47) one can read “…TT genotype of SNP rs3804100 (TLR2; Ser450Ser) was shown to be a protective factor against non-response”. Is that what the authors want to say here?

In conclusion, the present manuscript should be carefully revised both concerning English-language usage as well as concerning the interpretation of the results from the literature.

Experimental design

The review of the literature was well performed. Attention should be given to data interpretation.

Validity of the findings

As previously mentioned, authors should review the manuscript in order to avoid several confusing statements.

Reviewer 2 ·

Basic reporting

The authors responded sufficiently to my comments concerning the review's content.

I am still not fully satisfied with the English writing (some mistakes and typos are still present, see example below), but I would consider the manuscript with sufficient (not good) quality if the article is published in its current version.

Line 200 of PDF file: "mete analysis". Correct it in R2 or during proof (according to the Editorial decision).

Experimental design

The methods used for the selection of articles to be included in the review were significantly improved compared to the first version of the manuscript.

Validity of the findings

The new version of the manuscript presents the findings in a more organized way.

Additional comments

In general, the manuscript has been significantly improved compared to the first version presented by the authors. However, an additional grammatical revision would greatly improve the quality of the manuscript.

---

## Round 0.3 · Minor Revisions

Please address the remaining concerns of the reviewer and amend your manuscript accordingly.

Reviewer 1 ·

Basic reporting

This R2 is a truly improved version, although I would still suggest a few modifications.
First, the review of the English-language usage actually improved the manuscript, but a final revision by a fluent English speaker would eliminate some inadequacies. For example, at line 422, probably authors would like to say “According to in silico analysis” instead of “According to the silicon analysis”.
Also, authors present some opposing statements, and push too far some conclusions. For example, at line 522, they claim that “Clinicians may predict the clinical course of HCV-infected patients by TLR3 genotyping.”, which is an exaggerated conclusion based solely in association data, although at line 560, they acknowledge that “To date, the role of TLRs single nucleotide polymorphisms in HBV and HCV infection are still controversial.”

Experimental design

No comment.

Validity of the findings

No comment.

Additional comments

I suggest “Minor revision”, focusing in review of the English-language usage, without need to return the revised version to the reviewers, after these final corrections.

Reviewer 2 ·

Basic reporting

The manuscript underwent an English review, which resulted in some writing improvements. This version of the article shows sufficient quality for publication.

Experimental design

No additional comments beyond those already discussed in previous reviews.

Validity of the findings

No additional comments beyond those already discussed in previous reviews.

Additional comments

No additional comments.

---

## Round 0.4 · accepted · Accept

Thank you for addressing the remaining concerns of the reviewer. The revised version is acceptable now.